# Towards Understanding the Generalization Bias of Two Layer Convolutional Linear Classifiers with Gradient Descent

**Yifan Wu, Barnabás Póczos & Aarti Singh**
Machine Learning Department
Carnegie Mellon University
{yw4,bapoczos,aarti}@cs.cmu.edu

## Abstract

A major challenge in understanding the generalization of deep learning is to explain why (stochastic) gradient descent can exploit the network architecture to find solutions that have good generalization performance when using high capacity models. We find simple but realistic examples showing that this phenomenon exists even when learning linear classifiers — between two linear networks with the same capacity, the one with a convolutional layer can generalize better than the other when the data distribution has some underlying spatial structure. We argue that this difference results from a combination of the convolution architecture, data distribution and gradient descent, all of which are necessary to be included in a meaningful analysis. We provide a general analysis of the generalization performance as a function of data distribution and convolutional filter size, given gradient descent as the optimization algorithm, then interpret the results using concrete examples. Experimental results show that our analysis is able to explain what happens in our introduced examples.

## 1 Introduction

It has been shown that the capacities of successful deep neural networks are typically large enough such that they can fit random labelling of the inputs in a dataset (Zhang et al., 2016). Hence an important problem is to understand why gradient descent (and its variants) is able to find the solutions that generalize well on unseen data. Another key factor, besides gradient descent, in achieving good generalization performance in deep neural networks is architecture design with weight sharing (e.g. Convolutional Neural Networks (CNNs) (LeCun et al., 1998) and Long Short Term Memories (LSTMs) (Hochreiter & Schmidhuber, 1997) ). To the best of our knowledge, none of the existing work on analyzing the generalization bias of gradient descent takes these specific architectures into formal analysis. One may conjecture that the advantage of weight sharing is caused by reducing the network capacity compared with using fully connected layers without talking about gradient descent. However, as we will show later, there is a joint effect between network architectures and gradient descent on the generalization performance even if the model capacity remains unchanged. In this work we try to analyze the generalization bias of two layer CNNs together with gradient descent, as one of the initial steps towards understanding the generalization performance of deep learning in practice.

We introduce the following simple but realistic classification tasks on 1-D "images":

**Binary classification (Task-Cls):** Classify object A v.s. B.
$$x = [0, ......, 0, -1, 0, ..., 0] \rightarrow y = -1\,, \qquad x = [0, ..., 0, +1, 0, ......, 0] \rightarrow y = +1\,.$$

**First-person vision-based control (Task-1stCtrl):** Go to the proximity of object A. Left v.s. Right.
$$x = [0, ...1.., 0, ......, 0] \rightarrow y = -1\,, \qquad x = [0, ......, 0, ...1.., 0] \rightarrow y = +1\,.$$

**Third-person vision-based control (Task-3rdCtrl):** Control B to touch A. Left v.s. Right.
$$x = [0, ... + 1.... - 1....., 0] \rightarrow y = -1\,, \qquad x = [0, ...... - 1... + 1..., 0] \rightarrow y = +1\,.$$

One key property of all these three tasks we designed is that the data distribution is linearly separable. We compare a single layer linear classifier $\hat{y} = \text{sign}\left(w^T x\right)$ (Model-1-Layer) with a two layer convolutional linear classifier $\hat{y} = \text{sign}\left(w_2^T \text{Conv}(w_1, x)\right)$ (Model-Conv-$k$) where the convolution layer contains only one size-$k$ filter ($\text{output\_channel} = 1$) with $\text{stride} = 1$ and without non-linear activation functions. It is worth noting that Model-1-Layer and Model-Conv-$k$ represent exactly the same set of functions. Therefore, both of the two models have the same capacity and any difference in the generalization performance cannot be explained by the difference of capacity. Figure 1 shows that Model-Conv-$k$ outperforms Model-1-Layer on all of the three tasks. Our work is motivated by explaining the generalization behavior of Model-Conv-$k$ under gradient descent.

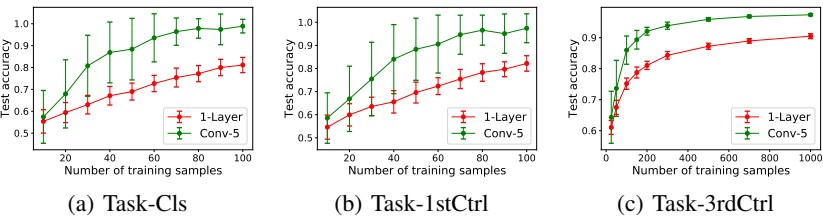

(a) Task-Cls  (b) Task-1stCtrl  (c) Task-3rdCtrl

Figure 1: Comparing the generalization performance between two models.

## 2 MAIN RESULTS

Explaining our empirical observations requires a generalization analysis that depends on data distribution, convolution structure and gradient descent. It can be shown that any of these three factors cannot be isolated from the analysis. We are the first to provide a formal generalization analysis that considers the interaction among data distribution, convolution, and gradient descent, which is necessary to provide meaningful understanding for deep networks. First we derive a closed form weight dynamics under gradient descent with a modified hinge loss and relate the generalization performance to the first singular vector pair of some matrix computed from the training samples. Then we interpret the results with one of our concrete examples using Perron-Frobenius Theorem for non-negative matrices. Our experiments show that our analysis is able to explain what happens in our examples.

We consider learning binary classifiers $\hat{y} = \text{sign}\left(f_w(x)\right)$ with a function class $f$ parameterized by $w$, in order to predict the real label $y \in \{-1, +1\}$ given an input $x \in \mathbb{R}^d$. A random label is predicted with equal chance if $f_w(x) = 0$. We denote the entire data distribution as $\mathcal{D}$, which one can sample data points $(x, y)$s from. Given a training set $D_{\text{tr}} \sim \mathcal{D}$ with $n_{\text{tr}}$ samples and a model $f_w$, we learn the classifier by minimizing the empirical hinge loss $\ell(w; x, y) = (1 - y f_w(x))_+$ using full-batch gradient descent with learning rate $\alpha > 0$. Given a classifier $f_w$ and data distribution $\mathcal{D}$, the generalization error can be written as $\mathcal{E}(w; \mathcal{D}) = \mathbb{E}_{(x,y) \sim \mathcal{D}}\left[\bar{\mathcal{E}}(y f_w(x))\right]$ where $\bar{\mathcal{E}}(x) = \mathbb{I}\{x < 0\} + \frac{1}{2}\mathbb{I}\{x = 0\}$. For the convenience of analysis, we use the following form to express Model-Conv-$k$: Let $A_x \in \mathbb{R}^{d \times k}$ be $[x, x_{\leftarrow_1}, ..., x_{\leftarrow_{k-1}}]$, where $k$ is the size of the filter and $x_{\leftarrow_l}$ is defined as the input vector left-shifted by $l$ positions. Then $f_w(x)$ can be written as $f_w(x) = w_1^T A_x^T w_2$. Further define $M_{x,y} = y A_x$ then we have $y f_w(x) = w_1^T M_{x,y}^T w_2$.

In our analysis we consider minimizing a modified version of the hinge loss $\ell(w; x, y) = -y f_w(x)$. We call it *the extreme hinge loss* because the gradient of this loss is the same as the gradient of the generalized hinge loss $\ell(w; x, y) = (\gamma - y f_w(x))_+$ with $\gamma \to +\infty$. Then the training loss becomes $\mathcal{L}(w; D_{\text{tr}}) = -w_1^T M_{\text{tr}}^T w_2$ where we define $M_{\text{tr}} = \frac{1}{n_{\text{tr}}} \sum_{(x,y) \in D} M_{x,y}$. The following Lemma shows that in full batch gradient descent $w_1^t$ and $w_2^t$ can be written in closed-forms:

**Lemma 1.** *Let $M_{\text{tr}} = U \Sigma V^T$ be (any of) its SVD such that $U \in \mathbb{R}^{d \times k}, \Sigma \in \mathbb{R}^{k \times k}, V \in \mathbb{R}^{k \times k}$, $U^T U = V^T V = V V^T = I$. Assume $w_2^0 = 0$. Then for any $t \geq 0$, $w_1^t = \frac{1}{2} V \Lambda^{+,t} V^T w_1^0$ and $w_2^t = \frac{1}{2} U \Lambda^{-,t} V^T w_1^0$ where we define $\Lambda^{+,t} = (I + \alpha \Sigma)^t + (I - \alpha \Sigma)^t$ and $\Lambda^{-,t} = (I + \alpha \Sigma)^t - (I - \alpha \Sigma)^t$.*

When $t \to \infty$ the weights converge to specific directions: $w_1^\infty \propto V_{:m} V_{:m}^T w_1^0$ and $w_2^\infty \propto U_{:m} V_{:m}^T w_1^0$, where $m$ is the number of singluar values that are the largest ($\sigma_1 = ... = \sigma_m$). We define the

*asymptotic generalization error* for Model-Conv-$k$ with gradient descent on data distribution $\mathcal{D}$ as $\mathcal{E}^\infty_{\text{Convk}}(\mathcal{D}) \doteq \mathbb{E}_{D_{\text{tr}},w^0_1}[\mathcal{E}(w^\infty, \mathcal{D})] = \mathbb{E}_{w^0_1,D_{\text{tr}},(x,y)}\left[\bar{\mathcal{E}}\left(w_1^{\infty T} M^T_{x,y} w^\infty_2\right)\right]$, which can be further upper bounded by $\mathcal{E}^\infty_{\text{Convk}}(\mathcal{D}) \leq \mathbb{E}_{D_{\text{tr}},(x,y)}\left[\bar{\mathcal{E}}\left(\min_{(u,v)\in UV^{M_{\text{tr}}}_1} v^T M^T_{x,y} u\right)\right]$ when $w^0_1 \sim \mathcal{N}(0, b^2 I_k)$, where $UV^M_1$ denote the set of left-right singular vector pairs corresponding to the largest singular value $\sigma_1$ for a given matrix $M$. When the first singular vector pair of $M_{\text{tr}}$ is unique, denoted by $(u, v)$, we have $m = 1$ thus $w^t_1$ converges to the same direction as $v$ while $w^t_2$ converges to the same direction as $u$. The asymptotic generalization performance is characterized by how many data points in the whole dataset can be correctly classified by Model-Conv-$k$ with the first singular vector pair of $M_{\text{tr}}$ as its weights.

We use our previously introduced task Task-Cls to show that our analysis is non-vacuous and able to explain the generalization advantage of Model-Conv-$k$ over Model-1-Layer. **Notation:** For any $l \in [d] = \{1, ..., d\}$ define $e_l \in \{0, 1\}^d$ to be the vector that has 1 in its $l$-th position and 0 elsewhere. Then the set of inputs $x$ in Task-Cls is the set of $e_l$ and $-e_l$ for all $l$. Let $\mathcal{U}[d]$ denote the uniform distribution over $[d]$. Given a training set $D_{\text{tr}}$ define $S_{\text{tr}} = \{l \in [d] : e_l \in D_{\text{tr}} \vee -e_l \in D_{\text{tr}}\}$ to be the set of non-zero positions that appear in $D_{\text{tr}}$. By applying the Perron-Frobenius theorem (Frobenius, 1912), the asymptotic generalization error can be decomposed into two parts:

**Theorem 2.** *Let $\Omega(A)$ be the event that $A$ is primitive and $\Omega^c(A)$ be its complement. Consider training Model-Conv-$k$ with gradient descent on Task-Cls. Then $\mathcal{E}^\infty_{\text{Convk}} \leq \Pr\left(\Omega^c(M^T_{\text{tr}} M_{\text{tr}})\right) + \frac{1}{2}\mathbb{E}_{l\sim\mathcal{U}[d]}\left[\Pr\left(\forall l' \in S_{\text{tr}}, |l' - l| \geq k\right)\right]$*

It can be shown that the first term is larger when the number of training samples $n_{\text{tr}}$ is very small while the second term becomes dominating when the number of training samples $n_{\text{tr}}$ is not too small. If we only consider the second term Model-Conv-$k$ requires approximately $2k-1$ times fewer samples than Model-1-Layer. We further show that, when comparing the sum of two terms with the generalization error of Model-1-Layer, the advantage exists only when $n_{\text{tr}}$ is not too small, which is well aligned with our empirical observation and also reveals an interesting fact that, unlike traditional regularization techniques, the generalization bias here requires a certain amount of training samples to be built up before saving the sample complexity effectively.

In our expriments we first show that $\mathcal{E}^\infty_{\text{Convk}}$ can be viewed as an upper confidence bound for the actual performance with the normal hinge loss (Figure 2(a)—2(c)). We then verify that the high variance with the normal hinge loss is caused by random initialization and good initializations under the normal hinge loss are the ones that converge faster to its limit direction under the extreme hinge loss (Figure 2(d) and 2(e)), which shows strong correlation between our analysis and what happens in our empirical observations.

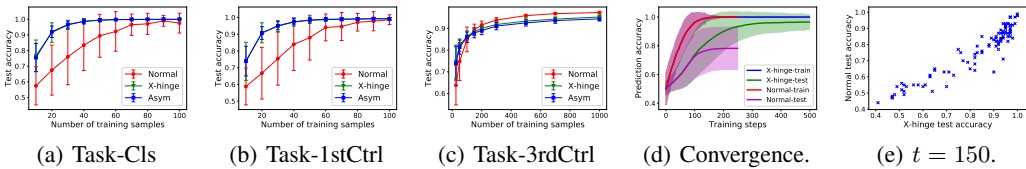

| (a) Task-Cls | (b) Task-1stCtrl | (c) Task-3rdCtrl | (d) Convergence. | (e) $t = 150$. |

Figure 2: Comparing estimated asymptotic error (Asym) v.s. finite time extreme hinge loss (X-hinge) v.s. normal hinge loss (Normal). The variance from weight initializations in Task-Cls.

## 3 CONCLUSION AND FUTURE WORK

We analyze the generalization performance of two layer convolutional linear classifiers trained with gradient descent. Our analysis is able to explain why, on some simple but realistic examples, adding a convolution layer can be more favorable than just using a single layer classifier even if the data is linearly separable. However, much work remains to be done:(i) Closing the gaps in normal hinge loss v.s. the extreme one as well as asymptotic analysis v.s. finite time analysis. (ii) How can we interpret the generalization bias as a prior knowledge. We conjecture that the jointly trained filter works as a data adaptive bias. (iii) Other interesting directions include studying the choice of $k$, making practical suggestions based on our analysis and bringing in more factors such as feature extraction, non-linearity and pooling.

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
