# OpenReview forum: "Towards Understanding the Generalization Bias of Two Layer Convolutional Linear Classifiers with Gradient Descent"
_ICLR.cc/2018/Workshop — Reject_

### Official Review · AnonReviewer3 · 2018-03-04
**an interesting analysis of deep learning combining network architecture, data distribution and gradient descent**

**Rating:** 7
**Confidence:** 3

**Review:**

In this paper, the authors contributed towards an understanding of deep learning by considering two layer convolutional linear classifiers trained with gradient descent. A feature of the analysis is that the authors considered a joint effect among network architecture, data distribution and gradient descent. This explains why two models with the same capacity may have different generalization ability. The results are interesting for us to understand the practical performance of deep learning.

Minor comments:
The description of 1-D "images" is not quite clear.
For Task-1stCtrl, do you mean instances x with only one '1' but the instances with y=-1 has '1' on the left-half, while that with y=1 has '1' on the right-half
For Task-3rdCtrl, do you mean instances with a single '+1'  and a single '-1' but the order of '+1' and '-1' are different for instances with y=1 and instances with y=-1?

---

### Official Review · AnonReviewer1 · 2018-03-09
**I'm not convinced that the artificial classification tasks are representative of real image-classification problems**

**Rating:** 5
**Confidence:** 2

**Review:**

The paper explores the difference in generalization performance between a linear model and a 1-layer convolutional model using three simple classification tasks, and theoretically and experimentally demonstrate that, on these tasks, the convolutional model generalizes better than the linear model.

I'm not persuaded that the three tasks are realistic. In the first, for example (Task-Cls), the examples are vectors with exactly one nonzero element, with this nonzero being either +1 or -1, which is also the label. The other two tasks are only slightly more complicated, and all three share the property that successful classification is about identifying one or two informative features, instead of aggregating information over the entire feature vector. Some classification tasks are like this, but I think that image classification is almost exactly the opposite: no one pixel, or handful of pixels, suffices to determine the label.

Theorems 1 and 2 demonstrate (with the simplifying assumption that a linear loss is being minimized) that the convolutional model generalizes better than the linear model. I wish that the authors had included an intuitive explanation of why this occurs, beyond the two theorems. It appears to me that, in the case of Task-Cls, the reason is that for the linear model there is only *one* informative feature per example, whereas inserting a width-k convolutional layer causes there to be k informative features, which is effectively reducing the dimension of the problem, and therefore improving generalization performance. Assuming that this intuition is correct, it appears to be a relic of the particular artificial problems under consideration, and is not something that can be easily used to draw conclusions about generalization performance of real models trained on real datasets.

---

### Decision · Program_Chairs · 2018-03-20
**ICLR 2018 Workshop Acceptance Decision**

**Decision:**

Reject

**Comment:**

Based on the reviews, this paper has not been accepted for presentation at the ICLR workshop. However, the conversation and updates can continue to appear here on OpenReview.